# PIK3CA Gene Mutations in HNSCC: Systematic Review and Correlations with HPV Status and Patient Survival

**DOI:** 10.3390/cancers14051286

**Published:** 2022-03-02

**Authors:** Daniela Cochicho, Susana Esteves, Miguel Rito, Fernanda Silva, Luís Martins, Pedro Montalvão, Mário Cunha, Miguel Magalhães, Rui M. Gil da Costa, Ana Felix

**Affiliations:** 1NOVA Medical School, NOVA University of Lisbon, 1099-085 Lisbon, Portugal; dcochicho@ipolisboa.min-saude.pt (D.C.); fernanda.silva@nms.unl.pt (F.S.); 2Virology Laboratory from Clinical Pathology Department, IPOLFG, 1099-023 Lisbon, Portugal; lmartins@ipolisboa.min-saude.pt (L.M.); mcunha@ipolisboa.min-saude.pt (M.C.); 3Clinical Research Unit, IPOLFG, 1099-023 Lisbon, Portugal; sesteves@ipolisboa.min-saude.pt; 4Pathology Department, IPOLFG, 1099-023 Lisbon, Portugal; mrito@ipolisboa.min-saude.pt; 5Otorhinolaryngology Department, IPOLFG, 1099-023 Lisbon, Portugal; pmontalvao@ipolisboa.min-saude.pt (P.M.); mmagalhaes@ipolisboa.min-saude.pt (M.M.); 6Post-Graduate Programme in Adult Health (PPGSAD), Morphology Department, University Hospital (HUUFMA), Federal University of Maranhão, São Luís 65080-805, Brazil; rmcosta@fe.up.pt; 7LEPABE, Laboratory for Process Engineering, Environment, Biotechnology and Energy, Faculty of Engineering, University of Porto, Rua Roberto Frias, 4200-465 Porto, Portugal; 8Centre for the Research and Technology of Agro-Environmental and Biological Sciences (CITAB), Inov4Agro, University of Trás-os-Montes e Alto Douro (UTAD), Quinta de Prados, 5000-801 Vila Real, Portugal; 9Molecular Oncology and Viral Pathology Group, Research Center of IPO Porto (CI-IPOP)/RISE@CI-IPOP (Health Research Network), Portuguese Oncology Institute of Porto (IPO Porto)/Porto Comprehensive Cancer Center (Porto.CCC), 4200-162 Porto, Portugal

**Keywords:** HNSCC, HPV, p16 INK4a, PIK3CA

## Abstract

**Simple Summary:**

Mutations of the PIK3CA gene are thought to contribute to the development of head and neck squamous cell carcinomas (HNSCC), especially those associated with human papillomavirus infection. Furthermore, these mutations may help identify patients who can benefit from specific targeted therapies. This study presents a systematic review of the PIK3CA mutations profile in HNSCC. The results are compared with a cohort of Portuguese patients to study the possible associations with HPV status and patient survival. The Portuguese cohort harboured PIK3CA mutations in 39% of patients, and there were no significant associations with the HPV status or with the OS. In this original case series, there was a statistically significant interaction effect between HPV status and PIK3CA mutation regarding disease-free survival. In HPV-positive patients, the PIK3CA wild-type is associated with a significant 4.64 times increase in the hazard of recurrence or death. Additional studies are needed to clarify the implications of PIK3CA mutations for patient prognosis.

**Abstract:**

PIK3CA mutations are believed to contribute to the pathogenesis of human papillomavirus (HPV)-associated head and neck squamous cell carcinomas (HNSCC). This study aims to establish the frequency of PIK3CA mutations in a Portuguese HNSCC cohort and to determine their association with the HPV status and patient survival. A meta-analysis of scientific literature also revealed widely different mutation rates in cohorts from different world regions and a trend towards improved prognosis among patients with PIK3CA mutations. DNA samples were available from 95 patients diagnosed with HNSCC at the Portuguese Institute of Oncology in Lisbon between 2010 and 2019. HPV status was established based on viral DNA detected using real-time PCR. The evaluation of PIK3CA gene mutations was performed by real-time PCR for four mutations (H1047L; E542K, E545K, and E545D). Thirty-seven cases were found to harbour PIK3CA mutations (39%), with the E545D mutation (73%) more frequently detected. There were no significant associations between the mutational status and HPV status (74% WT and 68% MUT were HPV (+); *p* = 0.489) or overall survival (OS) (3-year OS: WT 54% and MUT 65%; *p* = 0.090). HPV status was the only factor significantly associated with both OS and disease-free survival (DFS), with HPV (+) patients having consistently better outcomes (3-year OS: HPV (+) 65% and HPV (−) 36%; *p* = 0.007; DFS HPV (+) 83% and HPV (−) 43%; *p* = 0.001). There was a statistically significant interaction effect between HPV status and PIK3CA mutation regarding DFS (Interaction test: *p* = 0.026). In HPV (+) patients, PIK3CA wild-type is associated with a significant 4.64 times increase in the hazard of recurrence or death (HR = 4.64; 95% CI 1.02–20.99; *p* = 0.047). Overall, PIK3CA gene mutations are present in a large number of patients and may help define patient subsets who can benefit from therapies targeting the PI3K pathway. The systematic assessment of PIK3CA gene mutations in HNSCC patients will require further methodological standardisation.

## 1. Introduction

Head and neck carcinoma (HN) is the sixth leading cancer by incidence worldwide [1], according to data published in 2018, and is responsible for more than 800,000 new cases yearly and 450,000 deaths/year worldwide [2]. It comprises many different common and rare entities, the large majority (90%) being squamous cell carcinomas (SCC) [3]. There is a wide variation in disease progression and overall patient survival at different anatomic subsites [4,5], which may be explained by its multifactorial aetiology, in which environmental factors have a strong contribution, such as smoking and alcohol consumption. Infection with human papillomavirus (HPV), mainly the high-risk (HR) type HPV16, also plays an important role in a subgroup of these tumours, particularly in oropharyngeal cancers [6]. HPV-positive and HPV-negative tumours are clinically distinct [7], and two separate carcinogenesis routes are recognized [8]. Over the past decades, the incidence of HPV-associated oropharyngeal SCC has increased, as reported by the United States Center for Disease Control (CDC) in 2018 [9]. HPV-associated HNSCC has a substantially better prognosis after therapy compared with HPV-negative cases [10]. De-intensified therapy for these patients is being actively explored to reduce the associated morbidity while maintaining tumour control [11,12].

The genomic profile of HNSCC published by The Cancer Genome Atlas (TCGA) in 2015 allowed the development of new mutation studies and highlighted a high frequency of changes in components of the phosphatidylinositol-3-kinase (PI3K) signalling pathway, pointing out PIK3CA as the most frequently altered gene [13]. Activating PIK3CA gene mutations upregulate intracellular signalling via the PI3K–protein kinase T (Akt)–mammalian target of rapamycin (mTOR) pathway, contributing to multiple hallmarks of cancer, such as resisting cell death and uncontrolled proliferation [14,15,16]. As described previously, HPV genome integration leads to keratinocyte immortalization and transformation by inhibiting the tumour suppressor p53 and retinoblastoma protein (pRb) but also by interacting with other pathways, including PI3K/Akt/mTOR [17,18,19,20]. Several studies have addressed the role of the PIK3CA pathway in cervical cancer and other types of HPV-associated cancers [21,22,23,24,25]. In the evaluation of 151 HNSCC whole-exome sequences, the PI3K pathway was found to be the most commonly altered mitogenic pathway (30.5% of tumours) compared to the JAK/STAT pathway (9.3%) and MAPK pathway (8.0%) [26]. Among the PIK3CA mutations observed in HNSCC, 63% occur at three specific locations encoding the p110α subunit, namely E542, E545, and H1047, known as canonical mutations [27,28]. The PIK3CA mutation frequency among HPV-positive tumours was reported to be approximately half of that found in HPV-negative cancers [29]. Even so, the PIK3CA gene remains one of the most mutated in HPV-associated HNSCC [7,30]. Additionally, mutations have been associated with adverse outcomes in solid tumours, but their prognostic significance in oropharyngeal SCC is still unclear. In previous studies, the specific survival data did not differ between PIK3CA wild-type (WT) and mutated (MUT) lesions. However, WT-PIK3CA patients had significantly higher 3-year disease-free survival (DFS) compared to PIK3CA MUT patients, and a multivariable analysis for age, sex, smoking, TNM stage, and treatment showed associations between PIK3CA status and disease recurrence [31]. A greater understanding of therapies targeting the PI3K pathway has been achieved, developing new therapeutic combinations to improve the survival of HNSCC patients, but their efficacy is variable [32,33]. The identification of biomarkers to predict the response to therapy will allow the selection of patients who are more likely to respond to new therapeutic combinations [32], suggesting that different treatment strategies need to be considered based on the molecular phenotype of each tumour [34,35]. In this context, it is critical to refine the profile of PIK3CA mutations in HPV-positive and HPV-negative HNSCC patients and its association with prognosis.

To address these issues, the present study includes a systematic review of the frequency of PIK3CA mutations in HNSCC and the impact of the different detection techniques. The study also reports the prevalence of canonical PIK3CA gene mutations (substitutions for H1047L and E542K, E545K, and E545D) within an HNSCC Portuguese patient series at the time of diagnosis and conducted an exploratory analysis to test the hypotheses: (1) whether PIK3CA (canonical) mutations are associated with HPV-positive HNSCC, as evaluated by HPV DNA and p16INK4a and (2) whether PIK3CA mutations are independently associated with overall and disease-free survival.

## 2. Results

### 2.1. Systematic Review

We performed a systematic review to study the PIK3CA mutation frequency in different HNSCC cohorts (total case number range between 25 and 115 cases; total number of cases was 479) in scientific articles published from 2012 to 2021. The PIK3CA gene mutation analysis was performed by qPCR designed for specific targets in the regions known as hotspots of the PIK3CA gene exon 9 and exon 20, such E542K, E545K, E545D, H1047R, and H1047L, respectively. This analysis covers three different geographical regions: Europe (three studies), Asia (two studies), and North America (one study) (Appendix A).

The majority of the studies performed a mutational analysis using untreated tumour biopsy tissue, the male gender was the most represented in all studies, always representing more than half of the sample (55–93%), the mean age ranged between 63 and 65 years, and all the cohorts presented the active consumption of tobacco and/or alcohol. Almost all the studies reported a stratification by HN subsites (oral cavity, oropharynx, larynx, hypopharynx, and nasopharynx), and the majority of the cases originated from the oropharynx (five out of six studies, a total of 138 cases), followed by the oral cavity (four studies, a total of 97 cases), larynx (four studies, a total of 80 cases), hypopharynx (four studies, a total of 80 cases), and nasopharynx (three studies, a total 16 cases). All patients presented a histological diagnosis of SCC, most of them at advanced stages (III and IV). Only three of the six studies reported the HPV DNA status of the cohort. In total, 54 HNSCC HPV (+) and 219 HNSCC HPV (−) were evaluated. In three studies, the p16INK4 status was determined by immunohistochemistry with a total of 45 positive cases. The mutation frequency was estimated in a qualitative analysis of the number of cases with a mutation present. All studies reported PIK3CA gene mutations, and their frequency ranged from 8 to 32%. The lowest frequency (8%) was reported in a cohort of 113 cases (Borkowska 2021), mostly consisting of laryngeal tumours (43%) and, largely, HPV (−) lesions (77%). This low frequency of mutation detection can be related to the study’s design targeting only H1047R. Four other studies presented frequencies ranging between 16% and 18% and mainly consisted of lesions from the oropharynx and oral cavity. Among these four studies, HPV (−) HNSCC cases prevailed in the two studies where the HPV status was determined (García-Escudero 2018; SB Pattle 2017). All but one of these studies evaluated at least four mutations distributed both in the exon 9 and exon 20. The highest frequency was found (taking into account only (three cases) E545D (two cases) in exon 9 and H1047R (two cases) in exon 20). The highest reported frequency (32%) was observed in a Japanese study (Suda 2012) with a cohort of 115 cases, equally distributed across different subsites of the HN region (oral cavity *n* = 31, oropharynx *n* = 25, larynx *n* = 23, hypopharynx *n* = 25, and nasopharynx *n* = 11). Five mutations spots were evaluated in exon 9 and exon 20. The distribution of mutations by location revealed a predominance of oropharyngeal sites (oral cavity *n* = 9 (24%), oropharynx *n* = 10 (27%), larynx *n* = 8 (22%), hypopharynx *n* = 9 (24%), and nasopharynx *n* = 1 (3%) data from T Suda 2012).

In addition, we performed a second analysis to explore the HNSCC PIK3CA mutation profile in greater depth. For this second analysis, we assessed 17 selected articles comprising data from 1286 HNSCC patients published between 2006 and 2021 where mutations were detected using DNA sequencing techniques instead of PCR-based methods (Appendix A). The most used sequencing methodology was the classic Sanger assay (*n* = 10 studies), followed by cutting-edge NGS technology (*n* = 7). All studies evaluated tumour tissue, and the majority of samples were collected at the time of diagnostic (i.e., were treatment-naive samples). Primary tumours were located at different subsites from the HN region stratified by the oral cavity, oropharynx, larynx, hypopharynx, and nasopharynx; all data that did not specifically describe these locations were considered as not stratified. Three out of 17 studies evaluated SCC exclusively from the oral cavity, three others from the oropharynx, and one study from the hypopharynx. The remaining eight studies include data from different subsites. Overall, the most represented subsite was the oral cavity (*n* = 678 cases, reported by 11 articles), followed by the oropharynx (*n* = 340, reported by seven articles), larynx (*n* = 101, reported by five articles), hypopharynx (*n* = 94, reported by six articles), nasopharynx (*n* = 40, reported by five articles), and 32 non-stratified cases (reported by four articles). The HPV status was described in nine out of 17 studies: three studies evaluated single HPV status cohorts HPV (+) or HPV (−), and at least six studies presented both HPV (+) and HPV (−) cases. HPV (−) HNSCC was the most represented subgroup (*n* = 330 cases), followed by HNSCC HPV (+) (*n* = 244 cases). Overall, the frequency of PIK3CA mutations ranged between 3% and 36%. Only two studies reported the lowest frequency (3%) (Bruckman 2010 and Cortelazzi 2015), at least five studies presented frequencies above 20%, and three studies above 32%. Overall, 45 mutations were reported, distributed by both exon 9 and exon 20, and novel mutations were reported in a few studies. All mutations were stratified by exon, except for 30 cases reported in three studies.

The analysis based on general data for the mutational profile was supported by data reported in each of the studies. The three most frequently encountered mutations were the E545K mutation (*n* = 31 cases; data from 15 studies), followed by E542K (*n* = 21; nine studies) on exon 9 and H1047R (*n* = 16; nine studies), T1025T (*n* = 16; two studies), M10431 (*n* = 8; two studies), and G1049 (*n* = 2; three studies) on exon 20 (Appendix A). The PIK3CA-mutated cases per HN subsite were accurate; the highest number was in the oral cavity (*n* = 115, reported by seven studies), followed by oropharynx (*n* = 32, three studies), hypopharynx (*n* = 10, three studies), larynx (*n* = 8, three studies), and nasopharynx with no cases reported. We also evaluated the PIK3CA mutation (data reported from six studies) for subgroups of the HPV status. A total of 33 cases of HNSCC HPV (+) and 19 cases of HNSCC HPV (−) harboured PIK3CA mutations. In five of the 17 studies, the presence frequency of PIK3CA mutations was associated with the clinical parameters and correlated with the respective outcome (OS and DFS). Four studies estimated a better prognosis associated with the presence of the mutation (Alsofyania 2020, Lim 2019, García-Carracedo 2016, and Cohen 2011), and in one study, no correlation was found (Chau 2016).

### 2.2. Portuguese HNSCC Study Population

#### 2.2.1. Clinical and Demographic Characteristics

Cases were selected from a universe of 390 consecutive primary HNSCC cases located in the oropharynx or oral cavity at our tertiary cancer centre between 2010 and 2019. We excluded all cases with HPV-negative and missing information concerning the p16INK4a marker (*n* = 208) and all cases associated with HPV infection other than HPV16 (*n* = 79); the remaining 103 cases were included in the study and analysed for PIK3CA status (Appendix A). Eight cases were further excluded from the study due to technical failure in the PIK3CA mutational status evaluation. As such, our study sample comprised 95 patients. The demographic and clinical features of the excluded patients were similar to the included patients (Appendix A), and both groups showed overlapping survival curves (Appendix A), which suggests the absence of selection bias.

The demographic and clinical–pathological characteristics of the 95 HNSCC patients included in the analysis are summarized in Table 1. The average age at diagnosis was 62 years old and ranged between 37 and 90 years old. Most patients were men (73.7%, *n* = 70) and, according to self-reported consumer habits, the majority had active tobacco and/or alcohol consumption (70%, *n* = 66). Stratification was performed for each consumption habit. Alcohol exposure was characterised according to qualitative data in clinical records: never drank, ex-drinker, sporadic, moderate, and severe drinkers. The classification for tobacco consumption was defined as one pack/year (equal to one pack of cigarettes/day/year, with 20 cigarettes in a pack) distributed among current and heavy smokers. Current tobacco users included those who used tobacco ≤ 30 pack/years. Heavy users were those who smoked > 30 pack/years. The ex-smoker group included smokers who stopped until the date of diagnosis. Cases where the pack-a-year unit information was not available were classified as unrated smokers, and the data only reported consumption in a qualitatively way. Never users of tobacco or alcohol were defined as not having consumed either of these substances prior to cancer diagnosis. The analysis considered the two groups of active consumption vs. no consumption and did not consider the missing values. In 75 cases (79%), the primary tumour was located in the oropharynx, including the palatine tonsil (*n* = 47), the base of the tongue (*n* = 12), uvula (*n* = 2), soft palate (*n* = 10), trigone retromolar (*n* = 1), and oropharynx wall (*n* = 3). In the remaining 20 cases, the primary tumour was located in the oral cavity, including the tongue body (*n* = 13), buccal floor (*n* = 4), oral mucosa (*n* = 2), and mandible (*n* = 1). HPV infection was detected in 68 HNSCC cases, including 62 cases of HPV16 single-infection and six cases of coinfection with HPV16 and other HR or LR HPV types (HPV6, HPV18, HPV53, and HPV58). Data from p16^INK4a^ immunohistochemical assays were available for 89 patients. Patients were treated with different schemes involving surgery and radiotherapy (RT) as single therapies or in association with systemic antineoplastic treatment (Table 1). This treatment heterogeneity was expected, taking into consideration the clinical characteristics and tumour stage distribution observed in our cohort. Most patients showed complete response to the applied treatment (*n* = 66), but cases with disease persistence (*n* = 17) and relapse (*n* = 6) were also reported. In 12 cases, the treatment response could not be retrospectively evaluated (Table 1).

There were no differences between the wild-type (WT) and mutant (MUT) PIK3CA groups regarding age, primary tumour location, tumour stage at diagnosis, HPV16 infection status, and primary treatment administered (Table 1). The PIK3CA WT group had a numerically higher proportion of males (78% vs. 68%), p16 overexpression (43% vs. 30%), and complete responses to treatment (65% vs. 76%) compared to MUT PIK3CA, although none of these differences was statistically significant. There was a statistically significantly higher proportion of patients with active tobacco/alcoholic consumption habits among the WT compared to the MUT PIK3CA (76% vs. 60%) group; as such, this variable was also considered in the multivariable analysis of the prognostic impact of PIK3CA mutations.

#### 2.2.2. Prevalence of PIK3CA Mutations

The PIK3CA mutations were present in 39% of HNSCC cases (*n* = 37, 95% CI: 29–49%). The prevalence of PIK3CA mutations in HNSCC HPV (+) (*n* = 68) and HNSCC HPV (−) (*n* = 27) was, respectively, 36.8% (*n* = 25) and 44.4% (*n* = 12). Next, we analysed the distribution of PIK3CA mutations in patient subgroups defined by both the HPV DNA and p16^INK4a^ status, as summarised in Table 2. We could not demonstrate a statistically significant association between the presence of PIK3CA mutations and HPV DNA detection (*p* = 0.489, Table 1). When evaluating a possible association between PIK3CA mutation and the p16^INK4a^ status without stratification by the HPV DNA status, no statistically significant association was found (*p* = 0.163, Table 1). Similarly, there was no evidence of an association between the p16^INK4a^ status and PIK3CA mutation when considering HPV status stratification (*p* = 0.245).

#### 2.2.3. Classification of PIK3CA Gene Mutations: H1047L, E542K, E545K, and E545D

HNSCC cases were screened for four different single substitutions, known as canonical PIK3CA mutations. Among the cases harbouring PIK3CA mutations (*n* = 37), the majority carried E545D (73%; *n* = 27), followed by E545K (5%; *n* = 2), E542K (3%; *n* = 1), and H1047R (3%; *n* = 1). We observed the occurrence of combined substitutions E545D|E542K (5.4%; *n* = 2), E545D|E545K (8.1%; *n* = 3), and H1047R|E545D (2.7%; *n* = 1). The distribution of PIK3CA mutations in the HPV-positive and -negative subgroups is summarised in Table 3.

#### 2.2.4. PIK3CA Mutations and Patient Prognosis

The median follow-up determined using the reverse Kaplan–Meier method was 4.12 years (95% CI 3.1–5.5 years). A total of 46 deaths were reported during the follow-up period. The median overall survival (OS) in the whole sample (*n* = 95) was 4.75 years (95% CI 2.6–6.9 years). The 3-year OS was 58% (95% CI 48–70%). Disease-free survival (DFS) was evaluated in the 66 patients with a complete response to the first-line treatment (17 patients with persistent disease and 12 with missing information concerning response to treatment were excluded from the DFS analysis). Overall, the median DFS was 6.16 years (95% CI 4.6–NA), and the 3-year DFS was 75% (64–87%).

##### Analysis PIK3CA, HPV Status, and p16 Immunohistochemistry

OS and DFS were determined separately for patients grouped by the PIK3CA mutational status, HPV DNA, and p16^INK4a^ (Table 4 and Figure 1 and Figure 2). During the univariable analysis, we could not demonstrate a significant association between the PIK3CA mutation status and OS or DFS (Table 4 and Figure 1A,B). The HPV status was the only factor significantly associated with both OS and DFS, with patients with positive HPV having consistently better outcomes compared with the ones with non-detected HPV (Table 4 and Figure 2A,B). Patients with p16 overexpression had significantly longer DFS than patients with no p16 overexpression, but no significant difference could be demonstrated concerning OS (Table 4).

During the multivariable analysis adjusted for p16 overexpression, the HPV status, drinking/smoking habits, and age at diagnosis, the PIK3CA wild-type was associated with a two-fold increase in the hazards of death (HR = 2.15; 95% CI 0.98–4.73); nevertheless, this was not statistically significant at a 5% significance level (*p* = 0.056, Table 5). Therefore, we could not demonstrate an independent association of the PIK3CA mutation with the overall survival, controlling for p16 overexpression, HPV status, drinking/smoking habits, and age at diagnosis (Table 5). There was no evidence of a statistically significant interaction between the PIK3CA mutational status and HPV status in what concerns the OS (Interaction test: *p* = 0.7722, Figure 3A).

Regarding the association of PIK3CA mutation with DFS, the multivariable analysis showed a statistically significant interaction effect between HPV status and PIK3CA mutation (Interaction test: *p =* 0.026, Figure 3B) when controlling for p16 overexpression, drinking/smoking habits, and age at diagnosis. For this reason, we present the adjusted hazard ratios concerning the association of PIK3CA mutation with DFS in the groups of HPV-positive and not detected (Table 6). In patients HPV-positive, the PIK3CA wild-type was associated with a significant 4.64 times increase in the hazard of recurrence or death (HR = 4.64; 95% CI 1.02–20.99; *p =* 0.047). Conversely, no significant association could be demonstrated between the PIK3CA mutation status and disease-free survival in patients with nondetected HPV (HR = 0.38; 95% CI 0.06–2.24; *p =* 0.285).

## 3. Discussion

Several studies have pointed out the high frequency of PIK3CA mutations in HNSCC, especially in cases associated with HPV [30,36], and suggested that PIK3CA mutations have predictive and prognostic values [31,37]. Among the three canonical activating mutations, E542 and E545 target the helical domain of p110α, inducing PI3K hyperactivity by disrupting the regulatory activity of p85 on p110α [38]. Third, H1047 targets the p110α kinase domain and is thought to cause conformational change, allowing easier access to the phospholipid substrate [39]. Gene mutations targeting the p110α catalytic subunit were found in 56% of HPV-positive HNSCC tumours and in 34% of HPV-negative HNSCC tumours [7]. Moreover, data from a retrospective study with 87 oropharyngeal cancer samples reported that 10 out of 16 patients had activating mutations in the helical domain (E542K—8/16 and E545K—2/16) [36]. However, preliminary results from a clinical trial that stratified HNSCC patients by PIK3CA status showed no difference in the disease control rate over two months between patients with canonical PIK3CA mutations and those with WT-PIK3CA (36.4% vs. 38.9%). Thus, a deeper understanding of the prognostic value of PIK3CA mutations is needed [40]. To address this point, the present study evaluated the frequency of mutations in the PIK3CA gene in a Portuguese series with a diagnosis of SCC in the oropharynx and oral cavity and their association with HPV status and patient survival.

We also performed a systematic review to clarify the frequency of PIK3CA mutations reported in the scientific literature. We first analysed studies that employed qPCR to detect mutations, using a similar methodology to the one we used. Then, we performed a second and broader analysis dealing with studies that employed DNA sequencing methods, which allowed us to appraise the PIK3CA mutational spectrum within the HNSCC subsites in much greater detail. The first approach was not conclusive concerning the possible associations of PIK3CA mutations with HPV status and with patient prognosis, most likely due to the very limited number of studies involved. The second approach showed a higher frequency of PIK3CA mutations among the HPV-positive cases. Overall, 45 different mutations were reported, distributed both in exon 9 and exon 20, and novel mutations were reported, showing the potential of DNA sequencing techniques to refine the PIK3CA mutational landscape. HPV (+) patients were more likely to harbour mutations at E542 and E545, while HNSCC HPV (−) tumours showed more mutations at H1047, but the data remains insufficient to establish a pattern. Importantly, 4/17 studies reported improved OS or DFS for patients with PIK3CA mutations, even though most studies did not report any association, and another reported a lack of correlation between the mutational status and OS (Appendix A).

In the present study, PIK3CA mutations were found in 39% of our cases. This is the highest value among the previously published data (Borkowska 2021, García-Escudero 2018, Pattle 2017, Theurer 2016, McBride 2014, and Suda 2012). Since our patient cohort consisted of a majority of male patients with oropharyngeal, HPV-positive cancers, this result suggests a trend towards a high frequency of PIK3CA mutations among HPV (+) oropharyngeal tumours (Appendix A). Furthermore, these results are in accordance with the results of our systematic review, showing that PIK3CA is more commonly mutated in HPV-associated HNSCC. Nevertheless, we could not demonstrate a significant association between the mutation frequency and HPV status, as reported by some previous studies [7,36], which may be ascribed to different patient profiles in our cohorts.

Additionally, we could not demonstrate an association between PIK3CA and overall survival in the univariate or multivariable analyses, as previously reported by other research teams. Indeed, the multivariable analysis, where all the variables of interest, including p16^INK4a^ status and possible confounding factors such as age, smoking, and drinking consumption habits were taken into consideration, showed that the HPV DNA status was the only variable significantly associated with overall survival. This is in agreement with previous studies showing that HPV (−) patients have a poorer response to treatment and a worse prognosis compared with HPV (+) patients [10,11,12]. Additionally, this also suggests that it may be useful to first stratify patients by HPV DNA status and then study the prognostic impact of PIK3CA mutations within each group. In what regards DFS, the multivariable analysis showed a statistically significant interaction effect between HPV status and PIK3CA mutation when controlling for p16 overexpression, drinking/smoking habits, and age at diagnosis. For this reason, we presented the adjusted hazard ratios concerning the association of the PIK3CA mutation with DFS in the HPV-positive and HPV-negative groups. Importantly, in HPV-positive patients, wild-type PIK3CA was associated with a significant 4.64 times increase in the hazard of recurrence or death. This suggests that stratifying HNSCC patients on the basis of their HPV and PIK3CA status is a successful approach for predicting their prognosis more accurately and that the PIK3CA mutational status has an important prognostic impact when considered in the context of HPV. Numerous studies have associated PIK3CA mutations with a more aggressive phenotype [41,42]. However, contradictory results were obtained when evaluating different human cancers, and data showing a favourable impact on prognosis has been reported in patients with breast [43] and oesophageal carcinomas [44]. For the tumours of the HN region, the published data are still unclear, as shown in our systematic review. The present results contribute to elucidating this point, showing the importance of conjugating HPV and PIK3CA data to achieve a more closely tailored prognosis.

In the present data cohort, the most frequent mutation was E545D, followed by E545K and by E542K, all in exon 9, and one mutation in exon 20, H1047R. This is in agreement with the findings from our systematic review showing that the three most frequent mutations in previous reports targeted the same hotspots as in our cohort, namely E545K and E542K on exon 9 and H1047R, T1025T, M10431, and G1049 on exon 20. These data confirmed that exon 9 is the major mutations hotspot and that the oropharynx is the anatomic subsite most commonly affected. This is not a surprising finding, considering that HPV-positive tumours show higher mutation frequencies and are associated with this location. Taken together, these observations support the hypothesis that HPV-induced HN carcinogenesis preferentially involves PIK3CA mutations. In general, the data evaluated suggests that lesions with identical histological features may harbour different PIK3CA mutations, revealing a heterogeneous landscape among subsites of the HN region.

We concluded that PIK3CA alterations seem to be more common in HPV-positive HNSCC but were also common in HPV-negative patients, emphasizing the importance of this pathway independent of HPV infection [7,36]. This is similar to other SCC arising in other sites, such as vulvar carcinoma [45]. Therefore, determining which patients carry PIK3CA mutations is of great clinical significance, as it may help define patient subsets with specific needs and who may benefit from de-intensification treatments by promising targeted therapies against the signalling pathway involved. Specifically, this will allow the selection of patients who are more likely to respond to new therapeutic combinations [32], suggesting that different treatment strategies need to be considered based on the molecular phenotype of a tumour [34,35]. Stratifying patients by HPV DNA status is an effective approach to understand the prognostic impact of PIK3CA mutations, which may be masked when grouping HPV-positive and HPV-positive lesions together.

The main limitation of this study was the relatively small size of our patient cohort. The present results contribute to further defining the complex HNSCC landscape by determining the frequencies of canonical PIK3CA mutations in a Portuguese cohort subdivided into HPV-positive and HPV-negative patients and by showing the impact of PIK3CA mutations over DFS when patients are stratified according to their HPV DNA status.

## 4. Materials and Methods

### 4.1. Systematic Review of the Frequency of PIK3CA Mutations in HNSCC

The systematic review was performed according to the PICO model for estimating the frequency of PIK3CA mutations in HNSCC from previously published data. Population: HNSCC. Intervention: PIK3CA mutation frequency. Comparison: HNSCC HPV (+) vs. HNSCC HPV (−). The outcome it was not defined, since the expected results were not clear, not only because of the few studies that existed in the HN region but also because of the limiting case number on cohorts published. The inclusion criteria were established for the type of study (case series and case–control studies in humans); tumour sample type (FPPE and biopsies); tumour location (HN region oral cavity, oropharynx, larynx, hypopharynx, and nasopharynx); histological diagnosis (SCC); and molecular methodology performed (qPCR). It excluded all methodology based on RNA evaluation and the exclusive detection of other targets than PIK3CA reported. The search strategy contemplated the 3 standard databases on biomedicine: PubMed, Embase, and Cochrane, accessed in October and November 2021. The keywords PIK3CA and HNSCC were applied to a total of 374 articles, including 233 articles from PubMed, 154 articles from Embase, and 9 articles from Cochrane (Appendix A). Overall, 6 articles were selected (Appendix A). To further analyse more consistently and better clarify the mutational profile within the subsites HN region, our previously meta-analysis excluded 66 sequencing articles that may contribute to better translational knowledge about what type of PIK3CA mutation may occur and the oncogenic characteristics. It is generally expected that the frequency of estimated PIK3CA mutations may be different than those previously described, since the results were obtained from two different molecular approaches, such as qPCR detected at much lower compared to the higher sensitivity of the sequencing analysis [46]; additionally, the qPCR design targetable few specific regions contrasted with the whole-genome analysis allowed in sequence methods, adding a completely different perspective for the mutational landscape. For the evaluation of the HNSCC PIK3CA mutation profile, 66 sequencing articles eligible were validated following the exclusion criteria (1) the data from the RNA evaluation, (2) no data PIK3CA mutation profile, and (3) no more than 5 patient cases. We intend to define the somatic mutational profile HNSCC previously reported based on 17 articles evaluated (Appendix A).

### 4.2. Population

Archived paraffin-embedded primary tumour biopsy from 95 patients diagnosed and/or treated between 2010 and 2019 at the Head and Neck Surgery and the Otorhinolaryngology Departments from the Instituto Português de Oncologia (IPOLFG). Inclusion criteria were histological confirmation of primary HNSCC from the oropharynx and oral cavity and availability of materials for histological and molecular analyses. Patients with HNSCC associated with HPV infection other than HPV16 and with missing information concerning the p16 marker were excluded. All demographic, clinical–pathological, and follow-up data were retrieved from medical records.

### 4.3. Clinical–Pathological Characterisation

The tumours were categorised by histological type, grade, and TNM stage. A tissue microarray (TMA) was constructed for morphological evaluation, which was performed by two pathologists for diagnostic confirmation. Tumour samples of formalin-fixed paraffin-embedded tissue from biopsies or surgical specimens were stained with haematoxylin and eosin and p16 immunohistochemistry. Tissue microarrays were made with cores 1.5 mm in diameter retrieved from different areas of the tumour. Immunohistochemistry was performed on 4-μm-thick sections for anti-human p16 (clone E6H4, Cat. Number 805–4713, Roche Tissue Diagnostics, Pleasanton 94588 CA USA, prediluted for 4 min; pre-treatment ULTRA CC1-56 min, Ventana Medical Systems, 95050 Santa Clara, CA 95050 USA) following the manufacturer’s protocol, with appropriate positive and negative controls samples. Antigen detection was performed using the OptiView DAB IHC Detection Kit stained on the BenchMark ULTRA IHC/ISH Automatic staining platform (Ventana Medical Systems, USA), with diaminobenzidine as the chromogen to detect antigen expression. Tissue sections were counterstained with Mayer’s haematoxylin.

### 4.4. Identification of HPV Infection

HPV DNA detection was performed from tumour biopsies collected in ThinPrep conservative medium. The DNA was extracted using a spin silica filter column (QIAamp, a DNA Tissue Kit). Real-time PCR (qPCR) was performed by SYBR GREEN dye with specific L1 region primers (SPF10). The internal amplification control (albumin gene) was performed in independent reactions, along with HPV detection. A melting curve was performed to confirm the specificity (Tm 73 °C) of the 75-pb amplicon. The genotyping of the positive samples was processed by 2 different commercial kits: the LiRas assay (INNO-LiPA^®^ HPV Genotyping Extra II|Fujirebio, 19355 Malvern, PA, USA) detected 24 different HPV types and/or qPCR multiplex (Anyplex™ II HPV28 Detection|Seegene, 05548 Ogeum-ro, Songpa-gu, Seoul, Republic of Korea) able to detect 28 HPV types.

### 4.5. Mutation Analysis

DNA was isolated from biopsy samples of tumour tissue using QIAamp, a DNA Tissue Kit (Qiagen, Hilden, Germany). Detection of PIK3CA gene mutations (substitutions H1047L and E542K, E545K, and E545D) was conducted using commercial assays (AmoyDx^®^ PIK3CA Kit; TaqMan^®^ Mutation Detection Assays) based on a real-time PCR reaction. DNA isolated from PIK3CA gene mutation-positive cell lines (SW48 cell line: substitution E542K; MCF10A cell line: substitutions E545K and H1047R) was used as a positive control, and the internal control status amplified and detected a region of genomic DNA adjacent to the PIK3CA gene. Additionally, template control (NTC) was used to verify the positive contaminations. Reactions with 2.5 ng of DNA sample were run on a real-time PCR system (qPCR) using the universal mutation detection thermal cycling protocol. Amplification data were performed by qualitative analysis for the presence of each specific target according to the criteria described in the manufacturer’s instructions estimating the mutation status of each sample.

### 4.6. Statistical Analysis

We conducted a descriptive analysis for clinical and demographic characterisation of our cohort using absolute and relative frequencies for categorical variables, mean and standard deviation for quantitative variables, or median and interquartile ranges in the case of asymmetric distribution. The prevalence of PIK3CA mutations in the overall sample was calculated as a percentage with the respective exact binomial 95% Confidence Interval. The association between PIK3CA mutation and the demographic and clinical variables was tested using Pearson’s chi-square test or Fisher’s exact test for categorical variables, as appropriate, and the two-sample *t*-test for age. We also tested the association between PIK3CA mutation and p16 overexpression stratified by HPV status using the Cochrane–Mantel–Haenszel exact test (the Woolf test was used to check the homogeneity of the odds ratio across strata). We evaluated the prognostic impact of PIK3CA mutations and p16 overexpression on the overall survival (OS) and disease-free survival (DFS). Overall survival was defined as the period of time (in years) from the date of diagnosis to the date of death from any cause, patients alive were censored at the date of last follow-up assessment. Disease-free survival was evaluated in the patients with a complete response after the first treatment and was defined as the time from the end of the first treatment to disease relapse or death from any cause; patients alive without disease recurrence were censored at the date of the last follow-up assessment. We used Kaplan–Meier curves to visualize the differences in survival between subgroups defined by PIK3CA mutation, p16 overexpression, and HPV status and the log-rank test for group comparisons. Cox proportional hazards regression analysis was used to compute the hazard ratios (HRs) and 95% Confidence Intervals based on Wald statistics, with OS and DFS as the outcome variables and adjusting for age, HPV status, and drinking/smoking habits as potential confounding factors. Age and HPV status were chosen a priori as the most important confounding factors based on clinical criteria, and drinking/smoking habits were also included due to the imbalances observed between the PIK3CA MUT and WT groups in our cohort. The proportional hazards assumption was checked using statistical tests and graphical diagnosis based on Schoenfeld residuals. We also graphically assessed the functional form of the age variable in the models using Martingale residuals. As this analysis showed that the linearity assumption was not acceptable in the OS model, we adjusted the nonlinear effect of age with a smoothing spline using the “pspline” function of the R package “survival”. The potential interaction between HPV status and PIK3CA mutations on survival was tested using the log-likelihood ratio test between the fitted Cox regression models with and without the interaction term. The analyses were conducted in the complete case dataset. All statistical tests were two-sided, and we considered a significance level of 5%. As this was an exploratory study, no *p*-value correction for multiple testing was done. We used the software R package version 4.1.0 (http://www.R-project.org, accessed on 1 October 2021) [47].

## 5. Conclusions

Overall, these results confirmed a high frequency of canonical PIK3CA mutations (substitutions H1047L and E542K, E545K, and E545D) in HNSCC, including in HPV (−) cases, suggesting that screening for PIK3CA mutations should not be restricted to HPV (+) patients. Additional studies are needed to clarify the implications of PIK3CA mutations for HNSCC patient prognosis.

## Figures and Tables

**Figure 1 cancers-14-01286-f001:**
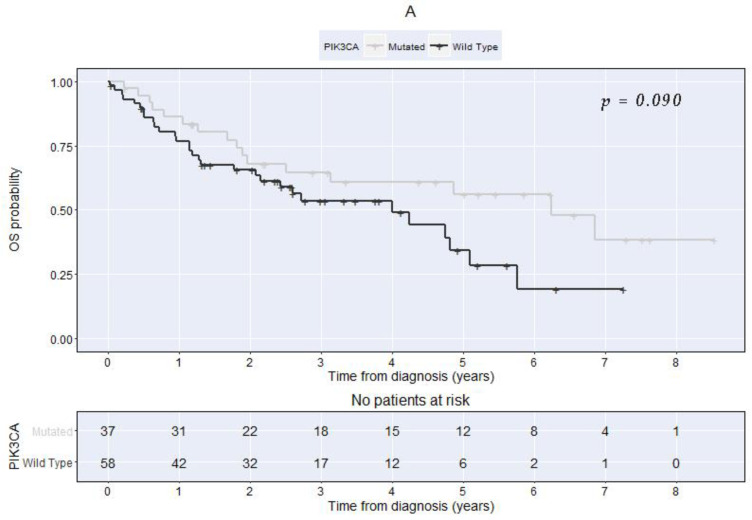
Kaplan–Meier curves comparing patients with PIK3CA-mutated vs. the wild-type at diagnosis (**A**). Overall survival (Log-rank test chi-square = 2.9 with 1 degree of freedom, *p* = 0.090). (**B**). Disease-free survival (Log-rank test chi-square = 1.7 with 1 degree of freedom, *p* = 0.198).

**Figure 2 cancers-14-01286-f002:**
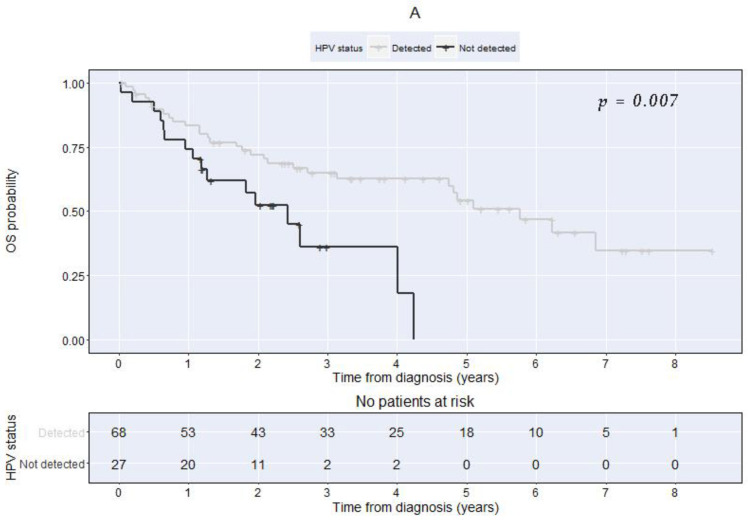
Kaplan–Meier curves comparing patients with detected vs. non-detected HPV at diagnosis. (**A**) Overall survival (Log-rank test chi-square = 7.3 with 1 degree of freedom, *p* = 0.007). (**B**) Disease-free survival (Log-rank test chi-square = 11 with 1 degree of freedom, *p* = 0.001).

**Figure 3 cancers-14-01286-f003:**
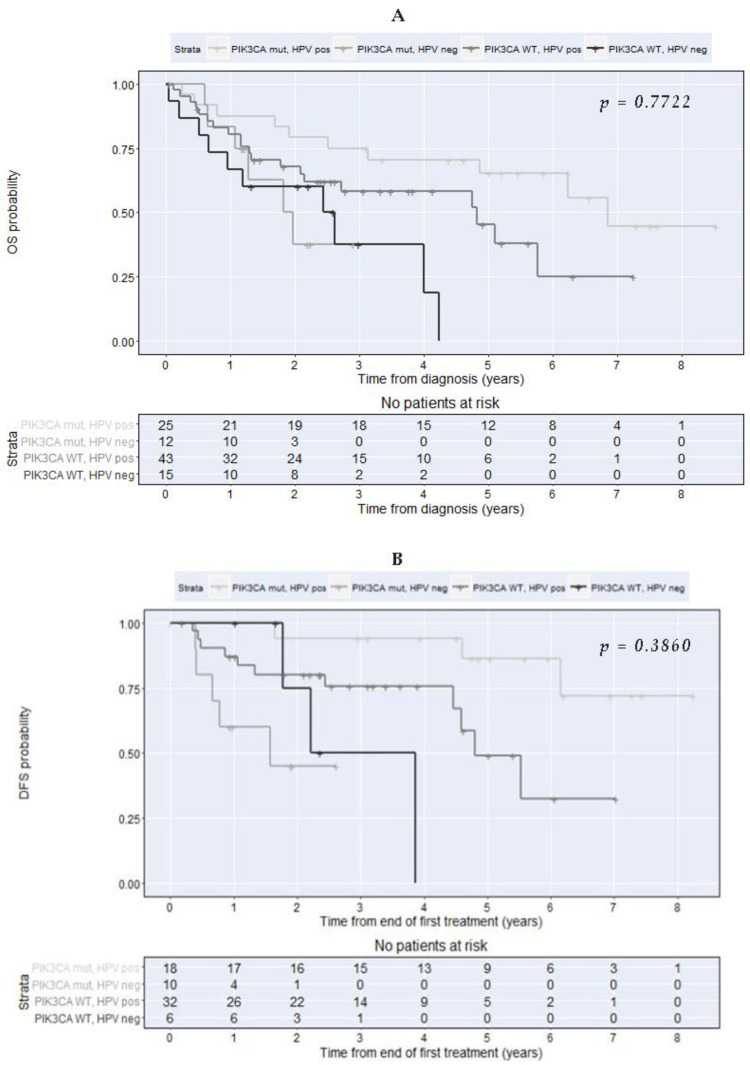
Kaplan–Meier plots for overall survival (**A**) and disease-free survival (**B**) by PIK3CA mutation and HPV status at diagnosis (*p =* 0.3860).

**Table 1 cancers-14-01286-t001:** Clinical, pathological, and demographic characteristics and association with PIK3CA mutation.

Variable Categories	PIK3CA Evaluation		Total Cases *n* = 95	*p*-Value
	WT (*n* = 58)	MUT (*n* = 37)		
	*n* (%)	*n* (%)	*n* (%)	
Gender				
Male	45 (77.6%)	25 (67.6%)	70 (73.7%)	0.2795
Female	13 (22.4%)	12 (32.4%)	25 (26.3%)	
Age at diagnosis				
Mean (Standard Deviation)	62 (10.7)	62 (12.7)	62 (11.5)	0.9740
≥65 Years	24 (41.4%)	12 (32.4%)	36 (37.9%)	0.3807
<65 Years	34 (58.6%)	25 (67.6%)	59 (62.1%)	
Consumption habits (tobacco and/or alcohol)				
Active consumption	44 (75.9%)	22 (59.5%)	66 (69.5%)	0.0325 *
Alcohol active	18 (31.0%)	10 (27%)	28 (29.5%)	
Tobacco active	7 (12.1%)	1 (2.7%)	8 (8.4%)	
Both active	19 (32.8%)	11 (29.7%)	30 (39.6%)	
No consumption	9 (15.5%)	13 (35.1%)	22 (23.2%)	
Never	5 (8.6%)	10 (27%)	15 (15.8%)	
Nonactive	4 (6.9%)	3 (8.1%)	7 (7.4%)	
No data	5 (8.6%)	2 (5.4%)	7 (7.4%)	
Tumour Anatomic region				
Oropharynx	46 (79.3%)	29 (78.4%)	75 (78.9%)	0.9135
Oral cavity	12 (20.7%)	8 (21.6%)	20 (21.1%)	
Tumour stage				
I	4 (6.9%)	4 (10.8%)	8 (8.4%)	0.9342
II	11 (19%)	8 (21.6%)	19 (20.0%)	
III	16 (27.6%)	10 (27.0%)	26 (27.4%)	
IV	25 (43.1%)	15 (40.5%)	40 (42.1%)	
No data	2 (3.4%)	0	2 (2.1%)	
p16 IHQ				
HNSCC p16(−)	29 (50.0%)	24 (64.9%)	53 (55.8%)	0.1627
HNSCC p16(+)	25 (43.1%)	11 (29.7%)	36 (37.9%)	
No data	4 (6.9%)	2 (5.4%)	6 (6.3%)	
HPV infection				
HPV16 DNA(−)	15 (25.9%)	12 (32.4%)	27(28.4%)	0.4887
HPV16 DNA(+)	43 (74.1%)	25 (67.6%)	68 (71.6%)	
Single infection	39 (90.7%)	23 (92.0%)	62 (91.2%)	
Co infection with other HR/LR HPV	4 (9.3%)	2 (8%)	6 (8.8%)	
Primary Treatment				
Surgery	1 (1.7%)	2 (5.4%)	3 (3.2%)	0.7376
Radiotherapy (RT)	10 (17.2%)	6 (16.2%)	16 (16.8%)	
Chemotherapy (CTX)	46 (79.3%)	29 (78.4%)	75 (78.9%)	
CTX + RT	26 (56.5%)	12 (41.4%)	38 (50.7%)	
Surgery + RT	9 (19.6%)	11 (37.9%)	20 (26.7%)	
Surgery + CTX	11 (23.9%)	6 (20.7%)	17(22.7%)	
No data	1 (1.7%)	0	1 (1.1%)	
Treatment response				
Complete	38 (65.5%)	28 (75.7%)	66 (69.5%)	0.5190
Persistence	11 (19.0%)	6 (16.2%)	17 (17.9%)	
No data	9 (15.5%)	3 (8.1%)	12 (12.6%)	

Legend: No data = no information available; RT = Radiotherapy; CTX = Chemotherapy; * *p*-value calculate for active vs. no consumption group.

**Table 2 cancers-14-01286-t002:** Frequency of PIK3CA mutations in HNSCC with and without HPV DNA and p16^INK4a^ overexpression (*n* = 89; p16 missing data *n* = 6).

HNSCC		PIK3CA Gene	
		WT *n* (%)	MUT *n* (%)
HPV (+)	p16(+)	22 (68.8%)	10 (31.3%)
	p16(−)	17 (56.7%)	13 (43.3%)
HPV (−)	p16(+)	3 (75%)	1 (25%)
	p16(−)	12 (52.2%)	11 (47.8%)

**Table 3 cancers-14-01286-t003:** Classification of PIK3CA gene mutations (substitutions H1047L and E542K, E545K, and E545D).

HPV Status	PIK3CA Gene MUT						
	One Substitution				Two Substitutions		
	E545D % (*n*)	E545K % (*n*)	E542K % (*n*)	H1047R % (*n*)	E545D|E545K % (*n*)	E545D|E542K % (*n*)	E545D|H1047R % (*n*)
HPV (+)	76% (19)	8% (2)	4% (1)	4% (1)	8% (2)	ND	ND
HPV (−)	67% (8)	ND	ND	ND	8% (1)	17% (2)	8% (1)
Total	73% (27)	5% (2)	3% (1)	3% (1)	8% (3)	5% (2)	3% (1)

Legend: ND = not detected.

**Table 4 cancers-14-01286-t004:** PIK3CA mutation, HPV, and p16 status association with the overall and disease-free survival by univariable analysis.

	Overall Survival				Disease-Free Survival			
	Median (Years)	3-Year % (95% CI)	HR (95%CI)	*p* *	Median, (Years)	3-Year % (95% CI)	HR (95% CI)	*p* *
PIK3CA								
MUT	6.2	65 (50–83)	1	0.090	NR	77 (63–95)	1	0.198
WT	4.0	54 (41–70)	1.71 (0.91–3.18)		4.8	72 (58–90)	1.80 (0.73–4.42)	
HPV status								
Positive	5.8	65 (54–78)	1	0.007	NR	83 (72–94)	1	0.001
Not detected	2.4	36 (19–69)	2.40 (1.15–4.61)		2.2	43 (21–90)	4.81 (1.74–13.29)	
p16 overexpression								
No	2.6	46 (33–64)	1	0.089	4.8	59 (44–80)	1	0.029
Yes	6.9	73 (60–90)	0.58 (0.31–1.10)		NR	91 (81–100)	0.34 (0.13–0.94)	

* Log-rank test; HR = Hazard Ratio; 95% CI = 95% Confidence Interval; NR = Not Reached; MUT = mutated; Wt = wild-type.

**Table 5 cancers-14-01286-t005:** Multivariable analysis of the PIK3CA mutation and p16 status association with overall survival, Cox regression model adjusted for HPV status, alcohol/tobacco consumption habits, and age at diagnosis.

Factor	Coef (SE)	HR (95%CI)	*p*-Value
PIK3CA			0.056
Mutated	1	1	
Wild-type	0.766 (0.401)	2.15 (0.98–4.73)	
p16 overexpression			0.670
No	1	1	
Yes	−0.189 (0.439)	0.83 (0.35–1.96)	
HPV status			0.016
Positive	1	1	
Not detected	1.069 (0.443)	2.91 (1.22–6.94)	
Consumption habits			0.040
Active	1	1	
No consumption	−1.041 (0.508)	0.35 (0.13–0.96)	
Age at diagnosis *			
Pspline, linear	0.045 (0.017)	---	0.008
Pspline, nonlinear	---	---	0.016

Coef = estimated coefficient from the model; SE = standard error of coefficient; HR = Hazard ratio; 95% CI = 95% Confidence Interval. * Smoothing splines was used to adjust for age with a smoothed curve without categorisation or linearity assumption (theta = 0.973, degrees of freedom for age term = 2.66).

**Table 6 cancers-14-01286-t006:** Multivariable analysis of PIK3CA mutation associated with disease-free survival, Cox regression model adjusted for p16 overexpression, alcohol/tobacco consumption habits, and age at diagnosis.

	Adjusted HR *	95% CI	*p*-Value
HPV status positive			
PIK3CA			
Mutated	1		
Wild-type	4.64	1.02–20.99	0.047
HPV not detected			
PIK3CA			
Mutated	1		
Wild-type	0.38	0.06–2.24	0.285

* Hazard ratio adjusted for p16 overexpression, consumption habits, and age at diagnosis.

## Data Availability

The data presented in this study are available on request from the corresponding author. The data are not publicly available due to privacy.

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
