# Peer review of "PIK3CA Gene Mutations in HNSCC: Systematic Review and Correlations with HPV Status and Patient Survival"

_cancers, 2022, doi:10.3390/cancers14051286_

Round 1

Reviewer 1 Report

The authors submitted a revised version with all my questions answered.

Author Response

Thank you for your appreciation

Reviewer 2 Report

I have been a reviewer on all rounds since the beginning. I am satisfied with the current version of the manuscript. Congratulations to the authors for successfully reviewing the manuscript and conducting the systematic review. The article is thus overall well rounded and a valuable contribution to science. Therefore, I recommend ACCEPT.

However, before the article is published, the following three minor points should be revised:
- Please mention Introduction first, then Material & Methods, then Results, then Conclusion.
- Please write out DFS in simple summary.
- The text in the Kaplan-Meier curves is very difficult to read. The text size has to be increased.

Author Response

However, before the article is published, the following three minor points should be revised:
Dear Reviewer

We are very thankful for all your critics that really improved our manuscript. 

  • Please mention Introduction first, then Material & Methods, then Results, then Conclusion.

We fully agree with you but we have a template done by Cancers MDPI that obliges to send the manuscript in that order. That is Material and Methods at the end.
- Please write out DFS in simple summary.

We did
- The text in the Kaplan-Meier curves is very difficult to read. The text size has to be increased.

We change the format in the word document, that was really difficult to read because the size of the all figure was small. The images on the supplement material are of good quality and can be easily read.

Thank you

Reviewer 3 Report

The authors have made adequate attempts to address all reviewers comments in this revised manuscript.

Author Response

We revised English language and style

This manuscript is a resubmission of an earlier submission. The following is a list of the peer review reports and author responses from that submission.

Round 1

Reviewer 1 Report

I thank the authors for allowing me to review their article "PIK3CA gene mutations in HNSCC: correlations with HPV status and patient survival" again after resubmission.

The work now offers essentially two contributions. First, the prevalence of different PIK3CA mutations in HNSCC of the oropharynx and oral cavity. Second, if applicable, there could be a negative effect in HPV+ patients who have wild-type PIK3CA. The authors conclude that therapy should be adjusted in this subpopulation.

The second statement is based on a putative HR of 4.64 (95% CI: 1.02-20.99). However, 1.02 is also in the range of possible. Therefore, the statement should be considered with caution. It is also illogical, since wild type occurs in 68.8% of the population of HPV+ cases (according to our own data). However, these percentages do not fit as an explanation for "However, a subset of patients will present late effects, in general, therapy still fails in 10% to 20% within these HPV-positive patients". Therefore, all very speculative on a very thin data basis.

That's why I had already recommended in my first review that you also do a meta-analysis to gather more data. The authors did not do this, although it is feasible.

Major notes:

  • Why is a distinction made between p16 overexpression and HPV+?
  • Why is age and consumption behavior not added as independent variables in the Cox regression in Table 4 as in Table 5?
  • Please perform a meta-analysis
  • Please shorten the text: Introduction has 1177 words, results 1328 words, Discussion 937 words, Material and Methods 1025, totaling just about 4500 words. This makes the manuscript very tenuous to read. It is not fun to read at all. I find there are far too many words for so few results. Partly, circumstances are described that are absolutely clear. The authors should drastically shorten their manuscript to ~3000-3500 words. Please!!! Likewise, I don't understand why you don't follow the IMRAD format (Introduction, Material & Methods, Results, Discussion) sequence.

Minor Notes:

- In the original submitted manuscript, the p-values were shown in the Kaplan-Meier curve. Please add again.

- The order in Table 1 takes some getting used to. I would first call WT, then Mutation and then Total.

- Why is the T stage not present in 2 cases?

- Why is the p16 IHQ status not present in 6 cases?

- Why is the therapy "without information" in one patient case? Please explain.

- Table 3: Replace HNSCC with Total and mention HPV-, then HPV+ and then Total.

Overall, I would like to mention that the manuscript has improved, but still not enough for a minior revision/accept. As a reviewer I have to make sure that the scientific quality is maintained, but at the same time I try to give you feedback to improve your manuscript enough to consider it for an Accept. I would like to encourage the authors to improve your article, please do a meta-analysis (they have the knowledge in R) and make your article more compact. Please less introduction, less material and methods, more critical discussion. In the Conclusion please more conservative statements.

Additionally my original review:

In the study by Cochicho et al, PIK3CA gene mutations in HNSCC were investigated in a retrospective cohort (2010-2019). The original cohort included 390 cases, of which only 95 cases were eventually included. In this study, oropharyngeal and oral cavity carcinomas were investigated together, although they differed in their mutation patterns to some extent. The main findings of the present study are that H1047L, E542K, E545K, and E545D are common mutations in OPSCC and OSCC in the examined cohort. PIK3CA frequently occur in HPV-negative patients and that a mutation in the PIK3CA gene is not associated with overall survival in the studied cohort. The strength of the work is that in the genetically very heterogeneous HNSCC, the PIK3CA mutation is investigated, which could potentially be important for a therapeutic approach targeting the PI3K pathway in HNSCC.

However, the work has several weaknesses. First, the small number of cases and the pooling of oropharyngeal and oral SCC. Also, far too much inductive statistics were performed. There are around 15-20 statistical tests and at the same time no correction for multiple testing was performed.

I doubt that the approach presented can answer the questions about the role of PIK3CA in OPSCC or OSCC to some extent.

However, since the paper could make a valuable contribution at its core, I recommend a thorough revision with a request for less inductive statistics and more descriptive statistics (again, confidence intervals with bootstrap), especially if only a handful of cases in certain groups are included for comparisons.

Additionally, for the manuscript to be acceptable in Cancers, a meta-analysis with inclusion of ALL previous data from publications on PIK3CA in HNSCC needs to be done and critically discussed. Likewise, more own cases should be included.

Additionally, therapeutic approaches and interactions with other mutations should be discussed for HNSCC. There must be a clear classification of the data also based on tumor location (oropharyngeal, oral, etc.). There should be less focus on HPV and more on PIK3CA.

My questions:

- Unfavorably, PIK3CA is also not associated with survival and at the same time PIK3CA could be a therapeutic target. This contradicts itself. How do you explain this contradiction?

- Why can't the p16 status be determined for the excluded cases? I recommend including more cases or performing a meta-analysis or both.

- Why have only 3 cases been treated surgically? (See Table 1) Or are the "Combine with Medication" also cases with surgery? HNSCC of the oral cavity are usually treated primarily with surgery. The tables should not be ambiguous.

- How was alcohol and tobacco status determined? These are often very weak data. Patients could also be active smokers for only 1 year. Is there any data available on Pack years? Or severity of alcohol consumption? If not, it is better to leave them out of the statistical analysis and describe them only descriptively. 

- Why is the T stage not present in 2 cases?

- Why is the p16 IHQ status not present in 6 cases?

- Age must not be categorized in the context of model calculations. 65 years is an arbitrary threshold.

- How were the confidence intervals calculated? I recommend a bootstrap approach (See bootstrap package in R, with 1000 repetitions).

Regarding the formatting and language of the manuscript:

- I would include material and methods before results.

- There are often multiple spaces between words.

Reviewer 2 Report

The sample size is very small and heterogeneous, so please add a supplemental table of all individual patients with all information analyzed in the manuscript.

Please discuss the following points: 

  1. A recent study declared the real-time PCR is not sensitive enough, which might explain why the rate of PIK3CA mutations was lower in this manuscript than previously reported data. (https://doi.org/10.3390/biom11060818)
  2. The sample size is not large, which restricted the sub-group analyses to compensate for multiple confounding factors: HPV status, oral/oropharyngeal, PIK3CA mutation, and so on.
  3. The amplification status of PIK3CA was not examined.
  4. The copy number of PIK3CA was not examined.